# Genetic Insights into Skeletal Malocclusion: The Role of the FBN3 *rs7351083* SNP in the Romanian Population

**DOI:** 10.3390/medicina60071061

**Published:** 2024-06-27

**Authors:** Adina Maria Topârcean, Arina Acatrinei, Ioana Rusu, Dana Feștilă, Radu Septimiu Câmpian, Beatrice Kelemen, Mircea Constantin Dinu Ghergie

**Affiliations:** 1Oral Rehabilitation Department, “Iuliu Hațieganu” University of Medicine and Pharmacy, Victor Babeș 8 Street, 400012 Cluj-Napoca, Romania; adinamtoparcean@gmail.com (A.M.T.); rcampian@gmail.com (R.S.C.); 2Molecular Biology Center, Interdisciplinary Research Institute on Bio-Nano-Sciences, Babeș-Bolyai University, Treboniu Laurian 42 Street, 400271 Cluj-Napoca, Romania; arina-lorena.acatrinei@usamvcluj.ro (A.A.); ioana.rusu@ubbcluj.ro (I.R.); beatrice.kelemen@ubbcluj.ro (B.K.); 3Doctoral School of Agricultural Engineering Sciences, University of Agricultural Sciences and Veterinary Medicine, Calea Mănăștur 3-5 Street, 400372 Cluj Napoca, Romania; 4Department of Molecular Biology and Biotechnology, Faculty of Biology and Geology, Babeș-Bolyai University, Republicii nr 44 Street, 400015 Cluj-Napoca, Romania; 5Department of Conservative Odontology, “Iuliu Hațieganu” University of Medicine and Pharmacy, “Iuliu Hatieganu” Victor Babeș 8 Street, 400012 Cluj-Napoca, Romania; mirceaghergie@yahoo.com

**Keywords:** malocclusion, cephalometric measurements, FBN3 gene, *rs7351083*

## Abstract

*Background and Objectives*: irregularities in the growth and development of the jawbones can lead to misalignments of maxillary and mandibular structures, a complex condition known as skeletal malocclusion, one of the most common oral health problems. Skeletal malocclusions, particularly Class II and Class III, can significantly affect facial appearance, chewing efficiency, speech, and overall oral health, often requiring orthodontic treatment or surgery to correct. These dentofacial anomalies are influenced by genetic and environmental factors and exhibit diverse phenotypic expressions. *Materials and Methods*: in this study, we investigated the correlation between the *rs7351083* SNP of the FBN3 gene that encodes a member of the fibrillin protein family and malocclusion risk in a group of 57 patients from Romania. *Results*: the results shed light on the relationship between the selected genetic marker and the investigated dentofacial disorder, revealing a positive association between the reference allele (A) and Class II and that the alternate allele (G) is associated with Class III. *Conclusions*: cephalometric analysis revealed no significant differences among genotypes, suggesting that while genetic factors are implicated in malocclusion, they may not directly affect cephalometric parameters or that the sample size was too small to detect these differences. The discovery of an A > T transversion in one individual with a Class II deformity underscores the genetic diversity within the population and the necessity of comprehensive genotyping to uncover rare genetic variants that might influence craniofacial development and the risk of malocclusion. This study highlights the need for larger studies to confirm these preliminary associations.

## 1. Introduction

Due to the high prevalence and treatment needs, and its correlation with altered function, malocclusion is considered by the WHO the third most important public oral health issue [1,2], after caries and periodontal disease.

Malocclusion is a developmental condition that affects the structural and functional relationship between maxillary and mandibular teeth, and is not considered a disease, but a natural variation of normal occlusion, which has been described since the 18th century [3]. The direct cause of skeletal malocclusion lies in the disruption of proper mandibular and/or maxillary growth during fetal development, and like many disorders of the head and face, malocclusions may appear as isolated phenotypes or as components of a syndrome [4]. A still incompletely understood combination of genetic, environmental, and ethnic factors that affect the various elements of the craniofacial complex, including cartilages, bones, teeth, and muscles, is etiologically related to the apparition of deviation from normal occlusion [2,5].

According to Angle’s classification, skeletal malocclusion can be divided into three groups: Class I, Class II, and Class III. The simplicity of Angle’s classification of malocclusion belies the fact that these dental malocclusions are not a single diagnostic entity [6,7]. Underlying this occlusion, conditions can be numerous skeletal and dental combinations. Diagnosis in orthodontics is rather a complex issue and starts with the history itself, whereby social history, socio-economic factors, and environmental factors can provide valuable clues to solve the problem. The assessment of malocclusion is based on clinical observations and the study of diagnostic records such as dental casts, facial photographs, and radiological records. Class I malocclusion describes a neutral relationship between the first maxillary and mandibular molars, with a normal maxilla and mandible relation resulting in a clinically flat profile.

Of interest to this paper are the last two groups. Class II skeletal malocclusion covers the phenotypic variation that results from a more protruded maxilla relative to the mandible or a retruded mandible relative to the maxilla, generating a clinically convex facial profile. Class II malocclusion is a commonly observed clinical problem, occurring in 19.56% of the global population, in the permanent dentition, with a higher prevalence of 22.9% in Caucasians. Regarding the permanent dentition, Europe showed the highest prevalence of Class II (33.51%) [2]. Analyzing the components of Class II malocclusion, McNamara [8] found that the maxillary skeletal position relative to the cranial base, most often, was in a neutral position (47–65%), in only a small percentage was it in a protrusive position (10–15%), and in 23–39%, it was in a retrusive position. Much emphasis has been placed on the size and position of the mandible relative to other craniofacial structures. Class II patients could be characterized as being deficient in mandibular size; other Class II patients have mandibles that are well formed but are retruded due to a posterior position of the glenoid fossa, resulting in a retrognathic relation to other craniofacial structures. The deficiency in the anterior–posterior position of the mandible is a common finding in Class II malocclusion, with about 60% of patients demonstrating mandibular skeletal retrusion [9]. Class II malocclusions display a distal relationship between the first maxillary and mandibular molars.

Class III skeletal malocclusion results from a more protruded mandible in relation to the maxilla or a retruded maxilla relative to the mandible, resulting in a clinically concave facial profile. Class III encompasses multiple types of phenotypic variation, with the most widely known being mandibular prognathism, in which the chin protrudes because of excessive mandibular growth in the sagittal plane [2,8,9,10,11]. In the permanent dentition, the prevalence of Class III is 5.93% [2], with a high prevalence among Mongoloids, mainly due to maxillary deficiency, in comparison with the persons with European ancestry, where Class III malocclusion is seen less frequently (0.8–4.2%) [10]. Class III skeletal malocclusion is not a single diagnostic entity and may have various combinations of skeletal and dental components. A mesial relationship between the first maxillary and mandibular molars characterizes Class III dental malocclusions. Mandibular skeletal protrusion, commonly cited as a major skeletal aberration in individuals with Class III malocclusion, combined with a normal anteroposterior position of the maxilla, was reported in less than 20% [12,13].

Class II and III are the most common malocclusions affecting orthodontic patients, having significant impacts not only on their masticatory functions, but also on their esthetic appearance and mental health, thus considerably impacting their quality of life [14,15,16,17]. Consequently, a deep understanding of the genetics underlying this complex trait is needed to assist orthodontists in properly diagnosing and treating malocclusion [18,19].

Both for Class II and Class III malocclusions, recent studies assessed correlations between phenotypes and the increasing number of polymorphic loci in candidate genes involved in the growth and development of the structures of the craniofacial complex (*EPB41*, *SSX21P*, *PLXNA*, *COL2A1*, *TGFB3*, and *LTBP2*, to name only few, for Class III [20], and *FGFR2*, *MSX1*, *MATN1*, *MYOH1*, *ACTN3*, *GHR*, *KAT6B*, *HDAC4*, and *AJUBA* for Class II [21]). Molecular pathways implicated in the development of bone (*TGFB3*, *LTBP*, *KAT6B*) and cartilage (*GHR*, *Matrilin-1*) have been identified as contributing factors to mandibular size discrepancies and Class III malocclusions characterized by variations in mandibular height and prognathism [5]. Nevertheless, these studies are plagued by small sample sizes, a lack of uniformity in experimental approaches, and focusing on populations with particular genealogies and geographic origin, aspects that make the results hard to extrapolate at a global level.

One gene previously identified as being associated with malocclusion is *FBN3* (fibrillin-3 precursor gene) [22]. Specifically, the *rs7351083* single-nucleotide polymorphism (SNP) (A allele) located within the gene’s regulatory region has been linked to an increased risk of Class III malocclusion [23]. This gene encodes proteins that constitute the extracellular matrix, which subsequently assemble to form microfibrils in various connective tissues. The expression of *FBN3* is notably upregulated during embryogenesis. [24].

The most current systematic review on the topic of genes and pathways associated with skeletal malocclusion identifies 19 genes associated with Class II malocclusion and 53 genes with Class III malocclusion, most of them enriched in pathways related to bone and cartilage regulation, but also to muscle [25]. The majority of SNPs’ associated with both Class II and Class III malocclusions are found in the introns of genes, probably causing the expression of protein isoforms of these genes more sensitive to factors of their surrounding microenvironment.

To this end, the present study aims to generate preliminary data regarding the extent to which the *rs7351083* SNP of the *FBN3* gene is correlated with the risk of developing malocclusion in the Romanian population, with regard to continuous phenotypic variation, as illustrated by four different cephalometric measurements. The findings from our study contribute to the expanding evidence implicating genetic variations in the *FBN3* gene in the development of malocclusion, notably mandibular prognathism. Additionally, the disparities in allele and genotype frequencies among different populations emphasize the significance of accounting for ethnic and demographic diversity in genetic investigations of orthodontic conditions.

## 2. Materials and Methods

This study was approved by the research and ethics committee of the University of Medicine and Pharmacy “Iuliu Hațieganu” Cluj-Napoca (number 97/3 August 2017). Written consent was obtained from all adults, or legal guardians in the case of minors, before they participated this study. The participants in this study were Caucasian patients of Romanian origin who sought orthodontic treatment at various private practices in three Transylvanian cities between 2016 and 2019.

### 2.1. Experimental and Control Groups

The experimental sample set consisted of 57 individuals, divided into three phenotypic categories: 18 with skeletal Class II malocclusion, 22 with skeletal Class III malocclusion, and the remaining 17 belonging to the control group.

### 2.2. Inclusion/Exclusion Criteria and Methodology for Skeletal and Dental Analysis

Control Group: the inclusion criteria for the control group mandated individuals exhibit Class I skeletal and dental relationships, well-balanced profiles, the absence of asymmetries, no shifts between centric relation (CR) and centric occlusion (CO), and no temporomandibular dysfunction symptoms.

Case Group: the subjects in case groups were classified as having Class II or Class III malocclusions based on clinical evaluation, facial photograph assessment, dental cast analysis, and cephalometric measurements.

Cephalometric Analysis: The cephalograms were analyzed using OnyxCeph software, incorporating the Steiner analysis, “Wits” appraisal, Bjorn–Jarabak analysis, and McNamara analysis. All cephalograms were taken in the natural head position.

Steiner Analysis: For the Steiner analysis, SNA, SNB, and ANB angles were measured. The reference points used for the Steiner analysis were S (*sella*) in the middle of the *sella turcica*, point A (*subspinale*) as the most posterior midline point on the maxillary concavity, and point B (*supramentale*) as the most posterior midline point in the mandibular concavity between the *infradentale* and *pogonion*.

SNA Angle: this assesses the maxillary position, with a mean of 82 ± 2°; values above this indicate maxillary prognathism, and those below indicate retrognathism.

SNB Angle: this reflects mandibular position relative to the cranial base, with a mean of 80 ± 2°; higher values suggest mandibular prognathism, and lower values suggest retrognathism.

ANB Angle: this indicates the maxilla–mandibular relationship, with a mean of 0–2°; values above this suggest Class II, and those below suggest Class III malocclusion [17].

Wits Appraisal: this addresses the limitations of the ANB angle, influenced by the length of the anterior cranial base, the vertical position of point N (*nasion*), and the rotation of the jaws relative to the anterior cranial base by measuring the linear distance from points A and B to the occlusal plane.

Ao-Bo Distance: With a mean of 0–2 mm, higher values indicate Class II malocclusion, while lower values indicate Class III. This measurement is influenced by the occlusal plane angle and alveolar bone dimensions.

#### Additional Measurements: Individualized ANB (Individual ANB): With a Mean Value of 0 ± 1, Values > 1 Indicate Class II, and Those <−1 Indicate Class III Malocclusion

McNamara Analysis: this measures the linear relationships between the midface (*condylion* to point A) and mandible length (*condylion* to *gnathion*), and the linear position of the maxilla and mandible relative to the cranial base.

Midface and mandible linear measurements help determine maxillary protrusion/retrusion and mandibular prognathism/retrognathism.

The linear distance from point A to the *nasion* perpendicular was measured. A distance of 0–1 mm is typical, with values above 1 mm indicating maxillary protrusion and values below 0 mm indicating maxillary retrusion. The mandible’s relationship to the cranial base was determined by measuring the distance from the pogonion to nasion perpendicular, where values between −4 mm and 2 mm are typical for Class II malocclusion. Measurements less than −4 mm suggest a retrognathic mandible, while values greater than 2 mm indicate a prognathic mandible. The anterior–posterior position of the mandible is also influenced by the angle between the anterior and posterior cranial base (N-S-Ar, or Saddle angle), with a mean vetric Comparisons Aalue of 123 ± 5°. Higher angles result in a more distal mandible displacement, producing a retrognathic profile, while lower angles result in a more forward mandible placement, leading to a Class III skeletal profile. The Articular angle (S-Ar-Go, mean of 143 ± 6°) indicates the relationship between the upper and lower parts of the posterior skeleton, with larger angles suggesting a retrognathic mandible and smaller angles indicating a prognathic mandible. Additionally, the length of the mandibular corpus should be equivalent to the anterior cranial base length by the age of 12. Lower values indicate a short mandible and a Class II skeletal profile, while higher values indicate a long mandible and a Class III profile.

Vertical Cephalometric Parameters: SN-MP Angle: With a mean of 32 ± 6°, high-angle patients typically have a distally positioned mandible, while low-angle patients have a more anteriorly positioned mandible. Additional angles such as FMA, OP, PP-MP, and PP-SN were evaluated to identify contributing vertical dimensions.

Study Focus: while a comprehensive cephalometric analysis was undertaken to provide detailed sample characterization and aid in treatment planning, this study concentrates on the results of the Steiner and Wits analyses concerning the genetic profile.

Patients with trauma, systematic diseases, asymmetries, and functional shifts were excluded from this study.

### 2.3. Molecular Investigation of FBN3 Polymorphism in Malocclusion Patients

#### 2.3.1. Sample Collection and DNA Isolation

Buccal swabs (Isohelix SK-1S/MS-01 Buccal Swabs, Cell Projects Ltd., Valletta, Malta) were collected following producer recommendations from all patients for molecular investigations and promptly transferred on ice to the genetic laboratory for DNA isolation and amplification. Genomic DNA was extracted from oral mucosa cells using a solution-based genomic DNA purification kit (Animal and Fungi DNA Preparation Kit, Jena Bioscience, Jena, Germany), following the manufacturer’s protocol. The quality and quantity of the genomic DNA were evaluated using a NanoDrop 1000 Spectrophotometer (ThermoFisher Scientific, Waltham, MA, USA). Subsequently, the gene fragment containing the target SNP (*rs7351083*) was amplified. A 364 bp segment of the FBN3 gene was amplified using specific primers designed with the Primer-Blast Tool (forward: 5′-GTCAGGAAGCGTGAGCCATTAT-3′; reverse: 5′-GCCAATCTCAGGCTTTCACTT-3′) [26].

#### 2.3.2. PCR Reaction Conditions and Thermal Cycling

The PCR reaction conditions were optimized to achieve consistent results, using the following reaction mixture: 1x Reaction Buffer, 2.5 mM MgCl2, 0.2 mM dNTP, 0.5 µM of each primer, 1.25 units/reaction of MangoTaq polymerase (Bioline, Meridian Bioscience, Cincinnati, OH, USA), and 1 µL of DNA template. The thermal cycling conditions consisted of an initial denaturation at 95 °C for 5 min, followed by 35 cycles of denaturation at 95 °C for 30 s, annealing at 62 °C for 30 s, extension at 72 °C for 30 s, and final extension at 72 °C for 5 min.

#### 2.3.3. Visualization and Purification of PCR Products

The PCR products were visualized on a 1.5% agarose electrophoresis gel (Cleaver Scientific Ltd., Warwickshire, UK). Successful amplifications were confirmed with clean negative controls, and the PCR products were subsequently purified from the agarose gel using the FavorPrep GEL/PCR Purification Kit (Favorgen Biotech Corp., Pingtung, Taiwan) and then sequenced by Sanger at Macrogen Europe (Amsterdam, The Netherlands).

#### Sequence Alignment and Identification of Polymorphisms

Alignment of the obtained sequences with the reference sequence (NC_000019.10) deposited in the NCBI database was performed using BioEdit Sequence Alignment Editor v. 7.2.5.0 [27] to screen for the *rs7351083* A > G/A > T *FBN3* polymorphism. All resulting DNA sequencing chromatograms were visually inspected to identify heterozygous positions.

#### 2.3.4. Statistical Analysis

Allele and genotype frequencies among individuals with malocclusion and control subjects were compared. The Pearson’s Chi-squared test with a simulated *p*-value based on 2000 replicates was utilized to examine the associations between FBN3 gene polymorphisms and the type of maxillary deficiency. Standardized residuals were depicted visually using the corrplot package in R 4.1.1 [28]. For the analysis of discrete cephalometric measures, the nonparametric Kruskal–Wallis test was employed to compare mean values between the genotypes. A significance level of *p* < 0.05 was considered statistically meaningful.

## 3. Results

### 3.1. Genetic Variant Identification

#### Sanger Sequencing and Allele Frequencies

We successfully conducted Sanger sequencing and identified genetic variants at the candidate locus (19p13.2, *rs7351083*) within the FBN3 gene for all 57 cases described in Table 1. The reference allele (A) was observed at a frequency of 55.26%, while the alternate variant (G) occurred at a frequency of 43.85%. The remaining percentage, accounting for 0.89%, corresponds to the presence of an A > T transversion detected in one heterozygous individual from the Class II malocclusion group. Interestingly, the frequency of the reference allele in the Romanian population appears to be notably higher than its average frequency in the European population (38.14%) and closer to that reported for African populations (56.16%), as documented in the ALFA project [29]. However, it is important to note that these differences may likely be attributed to the relatively small sample size of the investigated group. The AG heterozygous genotype emerged as the predominant genotype within our analyzed sample set, accounting for 64.91%. Following this, the AA homozygous genotype was observed at a proportion of 22.8%, while the GG homozygous genotype appeared at a frequency of 10.52%. Additionally, a small percentage (1.74%) of individuals exhibited the GT genotype.

### 3.2. Allelic Variability and Statistical Analysis

#### Allelic Frequencies across Malocclusion Categories

The allelic frequencies demonstrate variability among the three investigated orthodontic categories (Class II, Class III, and the control group), as illustrated in Figure 1A. The reference allele (A) is most prevalent in Class II malocclusion patients (61.11%), followed by 58.82% in the control group, and 47.72% in Class III malocclusion patients. Conversely, the alternate allele (G) exhibits a different pattern, being most frequent in Class III (52.27%) and least frequent in Class II (36.11%), with intermediate values in the control group (41.17%). Additionally, the mutant form T is present only in the Class II group, constituting a small percentage of 2.77%. The Pearson’s Chi-squared test with a simulated *p*-value based on 2000 replicates (χ-squared = 10.554, *p*-value = 0.01449) conducted for each allele within a patient category (Figure 1B) indicated a significant association between the reference allele and Class II malocclusion, with an even stronger association observed for the mutant allele T. Conversely, the mutant allele G exhibited an opposing pattern, displaying a negative association with this type of orthodontic deformity, with positive residuals (depicted in blue) detected solely in Class III malocclusion.

### 3.3. Genotypic Analysis

#### Genotypic Frequencies in Dentofacial Anomalies

The genotypic frequencies observed were not uniform across different types of dentofacial anomalies (Figure 2A). The AG heterozygote was the most abundant genotype, accounting for more than 50% in each category, whereas the AA homozygotes were detected in less than 30% in each analyzed group of dental occlusions. The GG genotype was identified only in Class III malocclusion (18.18%) and in the control group (11.76%), while the GT genotype appeared to be exclusively encountered in Class II malocclusion (5.55%). The correlation plot (Figure 2B) created based on the standardized residuals for the Pearson’s Chi-squared analysis (X-squared = 35.304, *p*-value = 0.0004998) shows a strong positive association between the GG genotype and mandibular prognathism, in contrast to its relationship with the Class II phenotype. All other genotypes were positively associated, to variable extents, with Class II skeletal malocclusion.

### 3.4. Cephalometric Parameter Analysis

#### Cephalometric Comparisons among Genotypes

Comparisons of the mean values of four cephalometric parameters (SNA, SNB, ANB, and Ao-Bo) among the different genotypes are visualized using box plots (Figure 3). The analysis reveals that there were no statistically significant differences in any of these cephalometric measurements between the genotypes. In other words, the variations in the genotypes did not correspond to meaningful differences in the cephalometric parameters measured.

## 4. Discussion

In this study, we successfully identified genetic variants at the candidate locus 19p13.2 within the *FBN3* gene (*rs7351083*) in a cohort of 57 individuals from the current territory of Romania. Our findings reveal intriguing patterns in allele and genotype frequencies that could have significant implications for understanding the genetic basis of malocclusion.

The identification of genetic factors and elucidation of their roles in mandible development are critical for accurately diagnosing orthodontic anomalies and facilitating the development of novel treatment approaches for these conditions [30]. Previous association studies, genome-wide association studies (GWASs), and genome-wide linkage analyses have identified numerous genes linked to mandibular prognathism, including *FBN3* [22]. However, the relationship between *FBN3* and malocclusion, particularly Class II malocclusion, has been less extensively studied compared to other candidate genes. Most research has focused on the association of *FBN3* with mandibular prognathism rather than its relation to Class II malocclusion. For instance, a systematic review and meta-analysis of genetic association studies [23] indicated that the presence of the A allele of the SNP *rs7351083* in *FBN3* increases the predisposition to Class III malocclusion.

In our study, the reference allele (A), observed at a frequency of 55.26% in the Romanian population, was significantly associated with Class II malocclusion, whereas the mutant allele (G) was positively associated with the Class III phenotype. This finding highlights a potential population-specific variation that might influence the prevalence and genetic study of orthodontic anomalies. Notably, the reference allele frequency in our Romanian cohort is higher than the average frequency reported for the European population (38.14%) and more aligned with that reported for African populations (56.16%), according to the ALFA project [29]. This discrepancy warrants further investigation to determine whether it reflects a true population-specific genetic variation or is an artifact of the relatively small sample size in our study.

An interesting incidental finding was the presence of a rare A > T transversion in one individual with a Class II deformity. This rare genetic variant underscores the genetic diversity within populations and the importance of comprehensive genotyping in genetic studies of orthodontic conditions. The detection of such rare variants can provide insights into the broader spectrum of genetic influences on craniofacial development and malocclusion.

Genome-wide linkage analysis in Korean and Japanese sibling pairs has previously identified 19p13.2 as a susceptibility locus for mandibular prognathism [31]. Conversely, an association study examining candidate polymorphisms involved in skeletal Class III malocclusion in a Brazilian population found that genes such as *MYO1H*, *GHR*, and *FGF10* contributed to maxilla–mandibular discrepancies rather than *FBN3*, despite testing four genetic markers of this gene, including *rs7351083* [32]. This difference in genotypic distribution between our Romanian sample and the Brazilian population highlights the complexity and heterogeneity of genetic contributions to malocclusion across different ethnic and demographic groups. For example, the GG homozygote genotype, strongly associated with Class III malocclusion in our cohort, was more frequent in the control group in the Brazilian study.

Further supporting the potential pathogenic role of *FBN3* in craniofacial development, Kalmari et al. [33] identified *FBN3*-*rs33967815* as a deleterious missense polymorphism potentially causing mandibular prognathism. It has been suggested that this SNP results in significant alterations in protein structure, potentially impacting the overall function of fibrillin-3 and, consequently, craniofacial morphology.

Although our study did not find statistically significant differences in cephalometric measurements among the genotypes, the median values for the four tested cephalometric landmarks were generally lowest in individuals with the GG genotype, except for the SNB angle. This outcome suggests that while there is likely a genetic component associated with malocclusion, this component might not directly influence the measured cephalometric parameters, or the sample size may be insufficient to detect subtle differences. The lack of significant differences could also denote that the cephalometric traits are influenced by a multitude of genetic and environmental factors, highlighting the complexity of craniofacial growth and development.

Future research should aim to verify and extend these findings by studying larger and more diverse populations, thereby verifying the observed associations and assessing their clinical applicability. Additionally, functional studies should be conducted to elucidate the specific biological mechanisms by which *FBN3* influences craniofacial development. For example, one recent study [34] considered the role of masseter muscle morphology and functional movements on the stomatognathic system and craniofacial form. Variations in gene expression within the masseter muscle, particularly involving Myosin Heavy Chain (MyHC) and other regulatory proteins, impact bone morphology due to the continuous force exerted on the mandible. This muscle is often studied in malocclusion research in order to assess expressions related to muscle regeneration and remodeling. Four fiber types in the masseter muscle (type I, type IIA/IIX, hybrid I/II, and neonatal) vary with malocclusion classifications, affecting bone modeling and facial growth. Understanding these mechanisms is crucial for translating genetic findings into clinical practice, potentially leading to more precise diagnostic tools and targeted therapeutic interventions for malocclusion and related craniofacial anomalies.

Our study adds to the growing body of evidence suggesting that genetic variants in the *FBN3* gene play a role in the etiology of malocclusion, in particular, mandibular prognathism. The population-specific differences observed in allele and genotype frequencies underscore the importance of considering ethnic and demographic diversity in genetic studies of orthodontic conditions. Although the cephalometric analysis did not reveal significant genotype–phenotype correlations in our sample, the potential genetic basis of malocclusion warrants further investigation through large-scale, multicenter studies and comprehensive functional analyses.

The comparative analysis with other populations, such as the Brazilian cohort, illustrated the heterogeneity and complexity of genetic contributions to malocclusion across different ethnic backgrounds. This emphasizes the need for population-specific genetic studies to better understand the prevalence and distribution of malocclusions, guiding more tailored diagnostic and treatment approaches.

Genome-wide linkage analyses and candidate gene association studies have identified several genetic loci and polymorphisms that contribute to craniofacial anomalies, including Class II and III malocclusions. These findings contribute to a more comprehensive genetic profile for orthodontic conditions and pave the way for personalized treatment strategies, potentially improving patient outcomes.

## 5. Conclusions

Our study provides valuable insights into the genetic underpinnings of skeletal malocclusions, specifically focusing on the role of the *FBN3 rs7351083* SNP in a Romanian cohort. We identified intriguing patterns in allele and genotype frequencies, highlighting a significant association between the reference allele (A) and Class II malocclusions, while the alternate allele (G) was found to be associated with Class III malocclusions.

The interesting discovery of an A > T transversion in one individual with a Class II deformity underscores the genetic diversity within the population and underscores the necessity of comprehensive genotyping to uncover rare genetic variants that might influence craniofacial development and the risk of malocclusion. This finding is a step toward unraveling the complex genetic architecture of skeletal malocclusions.

Our analysis did not reveal statistically significant differences in cephalometric measurements among different genotypes. This may suggest that while genetic factors play a crucial role in the etiology of malocclusion, their impact on cephalometric parameters might not be direct, or it might be too subtle to detect with the current sample size. The interplay of genetic and environmental factors in shaping craniofacial traits is likely to be complex and multifaceted.

However, the limitations of our study, including the small sample size and the inherent challenges in genetic studies of complex traits, highlight the need for larger, multicentric studies with diverse populations to validate and expand upon these findings. Further functional studies are essential to elucidate the biological mechanisms through which *FBN3* and other related genes influence craniofacial development.

Future research should aim to verify and extend these findings by studying larger and more diverse populations, thereby verifying the observed associations and assessing their clinical applicability. There is also a need for longitudinal studies to follow individuals over time to better understand the dynamic interplay of genetic and environmental factors in craniofacial development.

Functional studies should be undertaken to explore the specific pathways and mechanisms by which *FBN3* influences mandibular and maxillary growth. Advanced genomic technologies, including whole-genome sequencing and CRISPR-based gene editing, could be utilized to provide deeper insights into the role of *FBN3* and other candidate genes in craniofacial development.

The integration of genetic findings into clinical practice requires robust evidence-based guidelines. Therefore, collaborative efforts between geneticists, orthodontists, and researchers are necessary to translate genetic discoveries into precise diagnostic tools and effective therapeutic strategies for treating malocclusions and related craniofacial anomalies, a stage we have yet to reach.

In conclusion, our study underscores the significant role of genetic factors in the etiology of skeletal malocclusions and highlights the importance of considering population-specific genetic diversity in orthodontic research. The findings contribute to the growing body of knowledge aimed at improving the diagnosis, treatment, and prevention of malocclusions through personalized medicine.

## Figures and Tables

**Figure 1 medicina-60-01061-f001:**
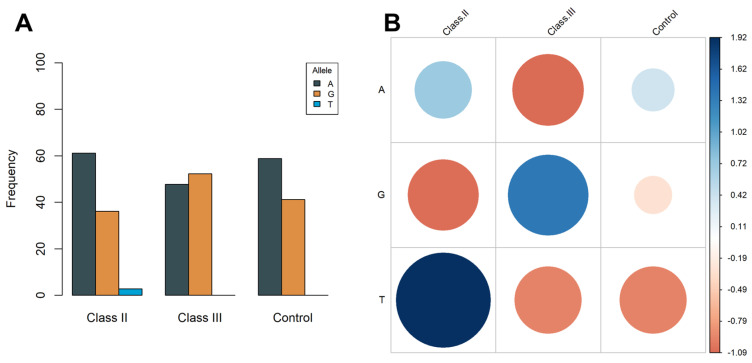
Allelic frequencies in the investigated malocclusion categories (**A**) and a correlation plot (**B**) for the Pearson’s Chi-squared test with the simulated *p*-value based on 2000 replicates for each allele in a class (χ-squared = 10.554, *p*-value = 0.01449).

**Figure 2 medicina-60-01061-f002:**
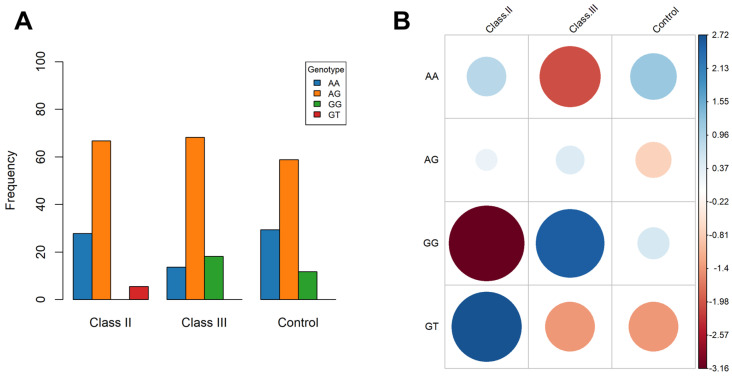
Genotypic frequencies in the investigated malocclusion categories (**A**) and a correlation plot (**B**) for the Pearson’s Chi-squared test with the simulated *p*-value based on 2000 replicates for each genotype in a class (X-squared = 35.304, *p*-value = 0.0004998).

**Figure 3 medicina-60-01061-f003:**
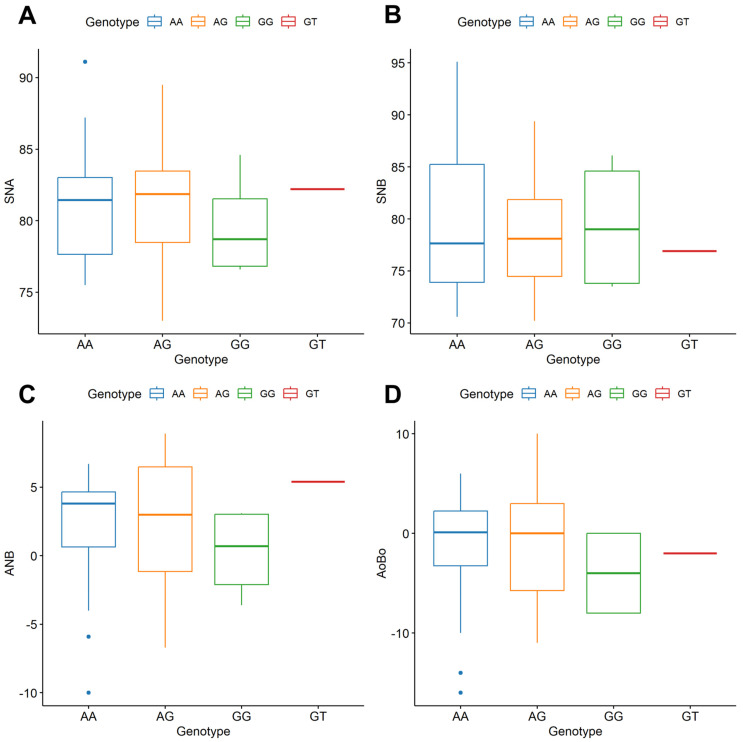
Comparison between mean values of four cephalometric measurements between genotypes ((**A**)—SNA; (**B**)—SNB; (**C**)—ANB; (**D**)—Ao-Bo). *p*-value of nonparametric Kruskal–Wallis test was higher than 0.05 in all instances.

**Table 1 medicina-60-01061-t001:** Description of investigated sample set.

Characteristic	Skeletal Class II (n = 18)	Skeletal Class III (n = 22)	Control Group (n = 17)
Mean age (SD)	19.2 (±7.59)	23.1 (±8.21)	22.1 (±8.69)
Sex_no. ind.			
Male	4	8	1
Female	11	9	10
Unknown	3	5	6
Measurements (°)(SD)			
SNA	80.91 (3.00)	81.25 (4.47)	80.99 (4.35)
SNB	74.57 (3.13)	83.74 (5.86)	77.4 (3.89)
ANB	6.39 (1.38)	−2.54 (3.39)	3.58 (1.70)
Ao-Bo	3.47 (3.20)	−7.52 (4.43)	0.14 (1.30)

## Data Availability

The data are contained within this article.

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
