# Peer review of "Genetic Insights into Skeletal Malocclusion: The Role of the FBN3 rs7351083 SNP in the Romanian Population"

_medicina, 2024, doi:10.3390/medicina60071061_

Round 1

Reviewer 1 Report

Comments and Suggestions for Authors

hello

thank you for an interesting paper

title is ok - main aim is described

abstract - well written, but not structured, please make it structured

key words - sufficient

introduction:

sufficient form

please improve the submission by adding features of normal class 1,2,3 dental/teeth malocclusion and the skeletal malocclusion, for example, skeletal class 3 malocclusion have some genetic links

material and methods;

the inclusion / exclusion criteria for the study are not written clearly - please improve them

material and methods section is long and unclear - please divide it into sub chapters and clearly define each step of this study

how was the swab and later genetic evaluation performed? techniques?

results section:

well presented figure section

results are somehow hard to understand, perhaps add sub sections for each issue?

discussion:

short and incocnlusive, requires improvements

discussion should be more representative

study limitations are missing 

Im suggesting to add top 5 key remarks

final conclusions, poorly written

please improve the paper, make it more structured and understandable for others

adding subsection would improve its understanding

also improve other issues marked above 

Author Response

     Dear Reviewer,

We appreciate the opportunity to submit the revised manuscript. Thank you for taking the time and effort to review it and provide helpful comments.

              Please see below, in blue, our responses given in a point-by-point manner. Changes to the manuscript are shown in underline, italic, blue text. These changes are also highlighted in the manuscript file by using the "Track Changes" function in Microsoft Word.

              We look forward to hearing from you regarding our submission.

              Sincerely,

              Prof. Dr. Dana FeÈ™tilă, PhD, DDS

Reviewer 1. Point 1:

abstract - well written, but not structured, please make it structured

  • Response 1-1: Thank you for your observation. Without changing the wording of the Abstract we structured it in 3 paragraphs:
  • In the first paragraph, the morphological changes associated with malocclusions are presented, as well as their detrimental effect on facial appearance, chewing efficiency, speech, and overall oral health.
  • In the second paragraph, the aim of this study and its results are synthetically presented
  • In the last paragraph, potential causes for the lack of association between chephalometric measurements and characterized genotypes are given.

Reviewer 1. Point 2:

Introduction: please improve the submission by adding features of normal class 1,2,3 dental/teeth malocclusion and the skeletal malocclusion, for example, skeletal class 3 malocclusion have some genetic links

  • Response 1-2: following clarifications were added as follows:
  • Lines 58-59: text was modified from “Class I skeletal malocclusion describes a normal maxilla and mandible relation with a clinically flat profile.” to “Class I malocclusion describes a neutral relationship between the first maxillary and mandibular molars, with a normal maxilla and mandible relation, resulting in a clinically flat profile.”
  • Lines 77-78: the following sentence was added: “Class II malocclusions display a distal relationship between the first maxillary and mandibular molars.”
  • Lines 88-89: the following sentence was added: “A mesial relationship between the first maxillary and mandibular molars characterizes class III dental malocclusions.”
  • Lines 104-107: text was added “Molecular pathways implicated in the development of bone (TGFB3, LTBP, KAT6B) and cartilage (GHR, Matrilin-1) have been identified as contributing factors to mandibular size discrepancies and Class III malocclusion characterized by variations in mandibular height and prognathism [5].” in order to underline the genetic links to class III malocclusion.
  • Lines 123-129: text was added “The most current systematic review on the topic of genes and pathways associated to skeletal malocclusion identifies 19 genes associated to class II malocclusion and 53 genes to class III malocclusion, most of them enriched in pathways related to bone and cartilage regulation, but also to muscle [25]. The majority of SNP’s associated to both class II and class III malocclusions are found in the introns of genes, probably causing the ex-pression of proteic isoforms of these genes more sensitive to factors of their surrounding microenvironment.”

Reviewer 1 Points 3 & 4 & 5:

Material and methods:

1-3 the inclusion/exclusion criteria for the study are not written clearly, please improve them

1-4 material and methods section is long and unclear – please divide it into sub chapters and clearly define each step of this study

1-5 how was the swab and later genetic evaluation performed? Techniques?

Response 1-3, 1-4, 1-5: The chapter “2. Materials and methods” was extensively rewritten and rearranged. It now starts at line 128 in the No Markup Track Changes format and ends at line 257. It is split in 3 sub-chapters: 2.1 Experimental and Control Groups, 2.2. Inclusion and Exclusion Criteria and Methodology for Skeletal and Dental Analysis and 2.3. Molecular Investigation of FBN3 Polymorphism in Malocclusion Patients. Inclusion and exclusion criteria are synthetically presented with key elements in bold. For sub-chapter 2.3. each step of the molecular analysis is treated in a separate paragraph previewed by a sub-title in bold. The clarification: “Buccal swabs (Isohelix SK-1S/MS-01 Buccal Swabs, Cell Projects Ltd., Malta) were collected following producer recommendations” was added at lines 287-288. All molecular methods and the evaluation of the rs7351083 are presented in 2.3. Molecular Investigation of FBN3 Polymorphism in Malocclusion Patients (Sample Collection and DNA Isolation; PCR Reaction Conditions and Thermal Cycling; Visualization and Purification of PCR Products; Sequence Alignment and Identification of Polymorphisms and Statistical Analysis.

Reviewer 1. Point 6: results are somehow hard to understand, perhaps add sub sections for each issue

Response 1-6: Results were structured starting at line 259 (No Markup Track Changes format), and ending at line 329) into sub-chapters: Genetic Variant Identification: Sanger sequencing and allele frequencies/Allelic variability and statistical analysis: Allelic Frequencies across malocclusion categories; Genotypic Analysis: Genotypic Frequencies in Dentofacial Anomalies; Cephalometric Parameter Analysis: Cephalometric Comparisons among genotypes.

Reviewer 1. Point 7 & 8:

Discussion: Point 7. short and inconclusive, requires improvements, discussion should be more representative, study limitations are missing, I’m suggesting to add top 5 key remarks

Point 8. Final conclusions poorly written

Response 1-7 and 1-8: The discussions underwent a revision to enhance the clarity and fluency of language (per Reviewer 2 request), with additional content incorporated to provide a more comprehensive exploration of the topic.

Study limitations were added to the Conclusion section: “However, the limitations of our study, including the small sample size and the inherent challenges in genetic studies of complex traits, highlight the need for larger, multicentric studies with diverse populations to validate and expand upon these findings. Further functional studies are essential to elucidate the biological mechanisms through which FBN3 and other related genes influence craniofacial development.” at lines 439 through 443 (No Markup Track Changes format). Conclusions were rephrased and extended as to underline some key remarks.

Reviewer 2 Report

Comments and Suggestions for Authors

The manuscript by Topârcean et al. provides genetic insights into some skeletal malocclusion. The paper is well-structured and interesting and can contribute to the scientific literature. Some suggestions for its improvement are listed below:

1. The introduction contains an extensive description of skeletal classes and less information about their genetics. The authors can summarize the first part and extend the second one.

2. A sentence explaining what makes this research innovative, unique, and significant for the scientific society should be added at the end of the Introduction section.

3. The sentence "Future research should aim to replicate these findings in larger and more diverse populations..." (line 317) must be edited. The future directions cannot be a replication of the research findings but improvement in its methodology.

Comments on the Quality of English Language

English editing is required.

Author Response

              June 10th, 2024

              To the editor and reviewers of manuscript medicina-3057992,

              We appreciate the opportunity to submit the revised manuscript. Thank you for taking the time and effort to review it and provide helpful comments.

              Please see below, in blue, our responses given in a point-by-point manner. Changes to the manuscript are shown in underline, italic, blue text. These changes are also highlighted in the manuscript file by using the "Track Changes" function in Microsoft Word.

              We look forward to hearing from you regarding our submission.

              Sincerely,

              Prof.Dr. Dana FeÈ™tilă, PhD, DDS

Reviewer 2:

Point 1. The introduction contains an extensive description of skeletal classes and less information about their genetics. The authors can summarize the first part and extend the second one.

  • Response 2-1. Introduction was completed with: Lines 104-107: text was added “Molecular pathways implicated in the development of bone (TGFB3, LTBP, KAT6B) and cartilage (GHR, Matrilin-1) have been identified as contributing factors to mandibular size discrepancies and Class III malocclusion characterized by variations in mandibular height and prognathism [5].” in order to underline the genetic links to class III malocclusion.
  • Lines 123-129: text was added “The most current systematic review on the topic of genes and pathways associated to skeletal malocclusion identifies 19 genes associated to class II malocclusion and 53 genes to class III malocclusion, most of them enriched in pathways related to bone and cartilage regulation, but also to muscle [25]. The majority of SNP’s associated to both class II and class III malocclusions are found in the introns of genes, probably causing the expression of protein isoforms of these genes more sensitive to factors of their surrounding microenvironment.”

Point 2. A sentence explaining what makes this research innovative, unique, and significant for the scientific society should be added at the end of the Introduction section.

Response 2-2. The following phrase was added at the end of the Introduction chapter: “The findings from our study contribute to the expanding evidence implicating genetic variations in the FBN3 gene in the development of malocclusion, notably mandibular prognathism. Additionally, the disparities in allele and genotype frequencies among different populations emphasize the significance of accounting for ethnic and demo-graphic diversity in genetic investigations of orthodontic conditions.” between lines 127-132 (No Mark-up Track Changes format).

Point 3. The sentence “Future research should aim to replicate these findings in larger and more diverse populations” (line 317) must be edited. The future directions cannot be a replication of the research findings but improvement in its methodology.

Response 2-3. The sentence was modified: “Future research should aim to verify and extend these findings by studying larger and more diverse populations, thereby verifying the observed associations and assessing their clinical applicability.” and is now located between lines 393-395 (No Mark-up Track Changes format).

The paper underwent a revision to enhance the clarity and fluency of language.

Round 2

Reviewer 1 Report

Comments and Suggestions for Authors

thank you for all changes

paper is suitable in current form